


# GCAM-CDR v1.0: Enhancing the Representation of Carbon Dioxide Removal Technologies and Policies in an Integrated Assessment Model

David R. Morrow[1], Raphael Apeaning[1], and Garrett Guard[1]

[1] Institute for Carbon Removal Law and Policy, American University, Washington, DC, 20016, USA

*Correspondence to*: David R. Morrow (morrow@american.edu)

**Abstract.** This paper introduces GCAM-CDR 1.0, an integrated assessment model for climate policy based on the open-source Global Change Analysis Model (GCAM). GCAM-CDR extends GCAM v5.4 by enabling users to model additional carbon dioxide removal (CDR) technologies and additional policies and controls related to CDR. New CDR technologies include terrestrial enhanced weathering with basalt, ocean liming, and additional versions of direct air capture. New CDR policies and controls

include integration of bioenergy with carbon capture and storage (BECCS) into the CDR market, interregional trade in CDR, exogenous control over the rate of growth of CDR, the ability to set independent targets for emissions abatement and CDR, and a variety of mechanisms for setting demand for CDR at the regional and/or global level. These extensions enhance users' ability to study the potential roles of CDR in climate policy.

## 1 Introduction

Carbon dioxide removal (CDR) involves capturing carbon dioxide ($CO_2$) from the atmosphere or ocean and storing or sequestering it for decades, centuries, or millennia. There are many different approaches to CDR, ranging from forest restoration and regenerative agriculture to direct air capture (DAC), bioenergy with carbon capture and storage (BECCS), and enhanced weathering (Fuss et al., 2018; National Academies of Science, Engineering, and Medicine, 2018). CDR has long played an important role in modeled climate mitigation pathways (Fuss et al., 2014), but until recently, most integrated assessment models

(IAMs) have only included a few approaches to CDR and offered limited flexibility with respect to policies related to CDR.

This paper describes GCAM-CDR 1.0, a variant of the Global Change Analysis Model (GCAM), whose purpose is to offer a wider variety of CDR technologies, policies, and controls. GCAM is an IAM developed by the Joint Global Change Resource Institute (JGCRI), which is a partnership between the University of Maryland and Pacific Northwest National Laboratory (PNNL) in the United States. GCAM-CDR was developed by the Institute for Carbon Removal Law and Policy (ICRLP) at

American University.

Section 2 describes GCAM-CDR. Section 3 presents some results from scenarios designed to compare the behavior of GCAM-CDR with GCAM 5.4 and demonstrate some of the new capabilities of GCAM-CDR. Section 4 offers some discussion of those results and the use of GCAM-CDR for studying the roles that CDR might play in climate policy.

## 2 Model Description

### 2.1 Overview of GCAM 5

GCAM-CDR 1.0 is based on GCAM 5.4 (Bond-Lamberty et al., 2021). Calvin et al. (2019) describe GCAM 5.1 in detail. Here we provide a brief overview of GCAM that covers essential context for understanding GCAM-CDR and important updates since version 5.1.





GCAM is a recursive-dynamic, partial-equilibrium IAM. It is used to study long-term climate change scenarios and

policies. The model contains five interconnected modules: climate, energy, water, land (including agriculture), and socioeconomics. The energy and socioeconomics modules represent 32 distinct geopolitical regions, while the water and land modules represent 384 land-water basins.

GCAM operates by seeking a set of market-clearing prices in each period for goods and services in the energy, land, and water modules, given exogenously specified socioeconomic inputs and policy constraints. This process is myopic, rather than

intertemporally optimizing: in each time-step, the model uses only information available at that time-step; it does not use information about future supply, demand, consumption, or policy. The end result is a vector of supply, demand, and price projections for each good and service for each period in each region of the model, as well as projections of emissions and global climate variables.

Users can simulate climate policies in GCAM by imposing exogenously specified prices on $CO_2$ emissions and/or non-

$CO_2$ greenhouse gas emissions, quantity-based constraints on those emissions, subsidies, taxes, renewable energy portfolio standards, and similar policies. Users can also use GCAM to identify a series of emissions taxes that achieves an exogenously specified climate target, such as limiting radiative forcing in 2100 to 1.9 $Wm^{-2}$.

GCAM has represented CDR technologies for over a decade. It has included bioenergy with carbon capture and storage (BECCS) technologies since at least version 3 (Luckow et al., 2010). BECCS has played an important role in deep carbonization

scenarios in GCAM, as in other IAMs (Fuss et al., 2014; Calvin et al., 2019; Köberle, 2019; Riahi et al., 2021). More recently, GCAM 5.4 has added direct air capture (DAC) technologies (Fuhrman et al., 2020, 2021; Bond-Lamberty et al., 2021). As discussed in Sect. 2.4, the options for modeling CDR policy in GCAM 5.4 remain somewhat limited. GCAM's default behavior is to drive the deployment of both BECCS and, when included, DAC by paying BECCS and DAC technologies at the price of carbon for each ton of carbon they remove.

## 55 2.2. Overview of GCAM-CDR

GCAM-CDR differs from GCAM in two main ways. First, it includes several new CDR technologies, bundled together in a CDR sector, all of which is achieved by adding XML input files. Second, it offers users a wider variety of policy options for shaping CDR markets and driving demand for CDR, which requires both new XML input files and changes to the source code. These changes are described in more detail in Sections 2.3 and 2.4, respectively.

In addition to these new features, the other main change is that GCAM-CDR slightly reorganizes parts of the energy system to more cleanly separate biofuels from fossil fuels. The purpose of this "bioseparation" is to enable better control over BECCS. GCAM 5.4 represents liquid and gaseous biofuels as becoming intermingled with fossil-based liquids and gas upstream in the energy sector. This allows for an elegant system of accounting for carbon flows and associated pricing (Kyle et al., 2011), but it creates challenges for tracking and managing BECCS. For example, the system allows gas-fired power plants with carbon

capture and storage (CCS) to transition gradually from low-emission technologies to negative-emission technologies by switching from fossil gas to biogas, but the amount of CDR achieved by such power plants is opaque to other parts of the model at runtime and therefore hard to control via policies that do not treat CDR and $CO_2$ emissions symmetrically. The bioseparation input files included with GCAM 5.4 keep bioliquids and biogas separate from their fossil-based counterparts upstream from potential BECCS applications, but allows them to intermingle as usual in downstream sectors, such as transportation, where CCS is not available.

This creates some deviations from GCAM 5.4's behavior, partly because of differences in the way GCAM parameterizes competition between technologies at different points in the energy system and partly because the bioseparation entails that biofuel-fired power plants and fossil fuel-fired power plants are distinct technologies, rather than technologies that can transition from one

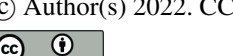



fuel to the other over time. The reorganization of the energy system is described in more detail in the model documentation, and the deviations introduced by the bioseparation files are quantified in Section 3.

Some minor changes to the source code, such as enabling regional tracking of $CO_2$ emissions independently of a carbon price, were also necessary to implement new policies and technologies. All changes to the source code are entirely backward compatible with GCAM 5.4: any input files that work with GCAM 5.4 will work with GCAM-CDR and will produce virtually indistinguishable results in the two models, as documented in Section 3. Changes to the source code are described in the model documentation and visible in the commit history of the code repository on GitHub, which was initially populated with the source

code for GCAM 5.4.

### 2.3. New CDR sector and technologies

GCAM-CDR 1.0 introduces several technologies whose primary purpose is to permanently remove $CO_2$ from the atmosphere. These technologies, which are grouped into a CDR supply sector, include two types of DAC; a terrestrial enhanced weathering (TEW) technology that involves spreading basalt on cropland; and ocean liming as an approach to ocean alkalinization or ocean

enhanced weathering (OEW). The DAC technologies in GCAM-CDR resemble but are not identical to those in GCAM 5.4. Fig. 1 depicts schematic representations of the default primary CDR technologies included in GCAM-CDR 1.0. Each technology produces an abstract good called CDR as an output, which can be interpreted as a certificate verifying the permanent removal of a unit of $CO_2$ from the atmosphere. The following subsections describe each technology in more detail, with key parameters given in the Supplemental Information.

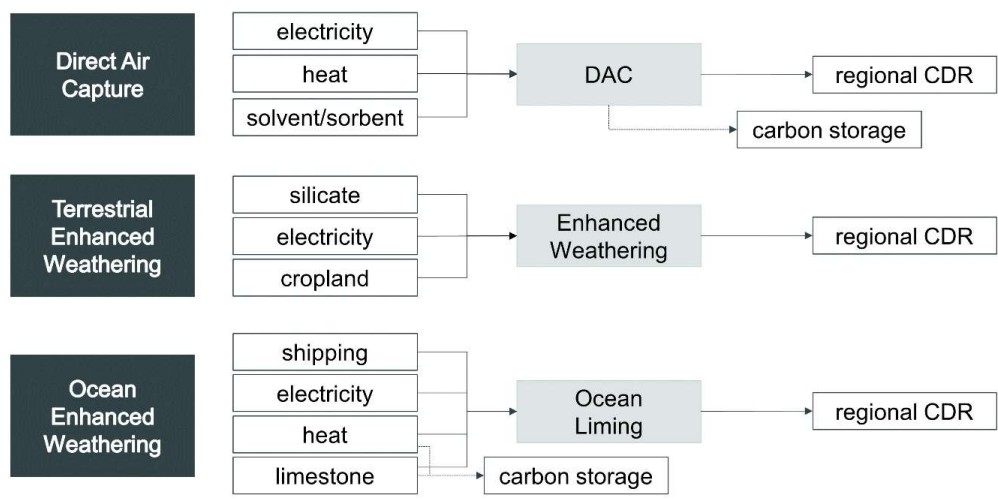


**Fig. 1. Schematic diagrams of new CDR technologies in GCAM-CDR 1.0. Direct air capture (DAC) technologies take electricity, heat, and either a solvent or sorbent as inputs, send $CO_2$ to geological storage, and produce an abstract good called CDR as an output. Terrestrial enhanced weathering requires silicate (basalt), electricity, and "cropland" as inputs and produces CDR as an output. Ocean liming requires limestone, electricity, heat, and international shipping as inputs,**
**sends process emissions to carbon storage, and produces CDR as an output. For more detailed diagrams related to each technology, see the Supplemental Information.**



To ensure that these new CDR technologies grow gradually, GCAM-CDR offers two main ways for users to constrain growth. The first is for users to specify demand for CDR in ways that grow gradually. The second is a flexible adaptation of the approach used to constrain growth in DAC in GCAM 5.4. In the latter approach, CDR technologies compete with a placeholder technology that does not remove any $CO_2$. The total output from CDR technologies is limited to a certain quantity in each period, set by the user, and the placeholder technology captures any excess CDR demand. Users can easily customize or eliminate this constraint on the growth of CDR technologies. Users can also choose whether to subject BECCS to the growth constraint or allow it to grow and operate independently of the new CDR technologies.

### 2.3.1. High-heat, liquid solvent-based DAC

GCAM-CDR represents a high-heat, liquid solvent-based approach to DAC, similar to the one developed by the Canadian company Carbon Engineering (Keith et al., 2018; National Academies of Science, Engineering, and Medicine, 2018). In our implementation, this technology takes natural gas and electricity as its main inputs and sends captured $CO_2$ to regional carbon storage markets, which allocate it to onshore or offshore geological storage. We assume an oxy-fired calciner that simultaneously captures atmospheric $CO_2$ and the emissions from the natural gas, sequestering both together.

This technology very closely resembles one of the DAC technologies introduced in GCAM 5.4, which is described by Fuhrman et al. (2020). Some parameters are based on work by Giulia Realmonte et al. (2019).

### 2.3.2. Low-heat, solid sorbent-based DAC

GCAM-CDR represents a low-heat, solid sorbent-based approach to DAC, similar to the one developed by the Swiss company Climeworks (Wurzbacher et al., 2016; National Academies of Science, Engineering, and Medicine, 2018; McQueen et al., 2020). In our implementation, this technology takes waste heat and electricity as its main inputs and sends captured $CO_2$ to the carbon storage market, which allocates it to onshore or offshore geological storage. The waste heat is modeled endogenously as a byproduct of thermal power plants and industrial energy use.

This technology loosely resembles one of the other DACCS technologies introduced in GCAM 5.4, but differs in that the GCAM-CDR version uses waste heat from other processes whereas the GCAM 5.4 version uses on-site process heat from natural gas. Some parameters are based on work by Giulia Realmonte et al. (2019).

### 2.3.3. Terrestrial enhanced weathering with basalt

GCAM-CDR represents an approach to terrestrial enhanced weathering that involves crushing basalt and spreading it on cropland, similar to the approaches described in several recent studies (Strefler et al., 2018; Beerling et al., 2020; Kelland et al., 2020). The default implementation of this technology takes basalt and electricity as inputs, as well as an abstract input called "cropland." The cropland input represents the limited supply of agricultural land on which to spread basalt, and it exhibits an upward-sloping supply curve to represent the increasing marginal cost of applying basalt to increasingly distant or hard-to-access areas. The $CO_2$ captured through TEW is assumed to be stored mainly in bicarbonate and carbonate minerals, but these are not explicitly modeled; removals are modeled simply as negative $CO_2$ emissions.

GCAM-CDR also includes an alternative, more experimental implementation that models basalt not as a standalone CDR technology, but as a type of fertilizer, based on studies suggesting that crushed basalt can increase agricultural yields (Nunes et al., 2014; Edwards et al., 2017; Beerling et al., 2020; Kelland et al., 2020). In this implementation, terrestrial enhanced weathering is excluded from the CDR sector and moved instead to the fertilizer sector. It operates as a partial substitute for nitrogen fertilizer, resulting in a mix of nitrogen fertilizer and basalt being applied to croplands. The modeled rate of application follows the



experiment described by Kelland et al. (2020), in which yield increases from basalt application approximately match the boost provided by nitrogen fertilizer in GCAM. In this implementation, endogenous limits on terrestrial enhanced weathering arise from the endogenous limits on fertilizer demand, which are based on the area of land under cultivation, demand for food, crop prices, etc. Further constraints arise from exogenously specified limits on the share of fertilizer demand that can be satisfied by basalt application.

The default implementation of enhanced weathering as a standalone CDR technology ignores the fertilization effects of basalt because the magnitude of that effect is still relatively unknown, having been documented only in a few case studies. We include the alternative implementation anyway to help users explore the agricultural and carbon implications of using basalt as a substitute for synthetic fertilizer, while acknowledging the large uncertainties around the fertilization effect.

  There are various other approaches to terrestrial enhanced weathering that are not included in GCAM-CDR 1.0. For instance, some
researchers have proposed using alkaline industrial wastes (Renforth, 2019) or more reactive minerals, such as dunite or olivine (Strefler et al., 2018). Others have proposed spreading crushed minerals in forests or unmanaged lands, rather than or in addition to croplands (Goll et al., 2021).

### 2.3.4. Ocean alkalinization with lime

  GCAM-CDR includes ocean liming as an approach to ocean alkalinization or ocean enhanced weathering, similar to approaches
described by Renforth et al. (2013) and Caserini et al. (2021). Ocean liming involves depositing lime into the open ocean, where it begins a short cascade of inorganic chemical reactions that lead to the ocean storing additional carbon as bicarbonates, allowing the ocean to absorb more $CO_2$ from the atmosphere while counteracting ocean acidification. Because of the modularity of the alkalinity calculations in GCAM's climate module, Hector, the beneficial effects of ocean liming on ocean alkalinity are not explicitly modeled in GCAM-CDR 1.0.

In GCAM-CDR, ocean liming uses natural gas, limestone, electricity, and an abstract good produced as a byproduct of international shipping. We assume an oxy-flash calcination process, as described by Renforth et al. (2013), that captures and sequesters the process emissions from both natural gas and limestone during calcination.

   The use of the international shipping byproduct serves both to endogenize the cost of depositing the lime in the ocean and to provide an endogenous upper limit to the amount of ocean liming that is feasible. Following Caserini et al. (2021), we assume
that lime would be spread from cargo ships, and that ocean liming cannot use more than 13% of global shipping capacity, although total capacity does increase endogenously in response to demand from the ocean liming industry. See the Supplemental Information for details.

   There are various other approaches to ocean alkalinization that are not included in GCAM-CDR 1.0. For example, some researchers have proposed dumping silicate minerals into the ocean (Köhler et al., 2013), spreading crushed silicates on beaches
or coastal seabeds (Meysman and Montserrat, 2017), exposing carbonate minerals to seawater in an artificial, high-$CO_2$ environment  (Boyd et al. 2019), releasing carbonate and bicarbonate produced as a by-product of hydrogen production (Rau et al., 2018), or electrochemically separating seawater and pumping acidity into deep water (Tyka et al., 2022).

### 2.4. CDR policy options

  Although it is not normally described this way, GCAM 5.4 implicitly implements a very specific CDR policy: any technology that
removes a ton of carbon from the atmosphere receives a subsidy equal to the carbon price, which has to be repaid if and only if the carbon is re-emitted downstream. (Earlier versions of GCAM work similarly.) GCAM models raw biofuels (e.g., corn for ethanol or wood pellets) as removing carbon from the atmosphere, thus earning a subsidy. (The carbon implications of growing and





harvesting the biomass are handled separately by GCAM's land-use module.) If those biofuels are consumed without CCS, the carbon is re-emitted and the subsidy is repaid, but technologies that capture and sequester the carbon in those biofuels effectively

"keep" the original subsidy for themselves. Thus, BECCS technologies end up receiving a subsidy, which is equal to the carbon price, for each ton of (biologically derived) carbon they sequester. Furthermore, in the default set-up for GCAM 5.4, it is implicitly assumed that this subsidy comes out of revenues from a carbon price, and that the total available subsidy for CDR can only exceed revenues from the carbon price by a fixed amount, referred to as the "negative emissions budget." CDR services can be traded between regions if and only if the regions are subject to a shared greenhouse gas constraint policy, such as a quantitative limit on

global $CO_2$ emissions, in which case negative emissions in one region can offset emissions in another region.

In the real world, each part of this system would have to result from specific choices in the design of carbon pricing and climate policy. Yet in GCAM, the model behavior can seem like a natural consequence of pricing GHG emissions, rather than a specific policy that connects demand for CDR to a price on emissions in a particular way. This is especially easy to overlook because much of this behavior is either set in the source code or emerges from more general carbon accounting mechanisms in

GCAM, rather than any policy explicitly represented in the model inputs.

GCAM 5.4 adds the ability to exogenously specify maximum demand for DAC, with actual demand responding endogenously to the carbon price. When users model $CO_2$ or GHG constraints or certain kinds of climate policy targets, however, GCAM 5.4 implicitly models any demand for DAC as an offset for other emissions: setting DAC to remove $n$ metric tons of $CO_2$ from the atmosphere permits an additional $(n - k)$ tons of GHG emissions, measured in $CO_2$eq, where $k$ equals gross emissions

from the energy inputs to DAC. This builds in the kind of "moral hazard" or "mitigation deterrence" that is frequently raised as a concern in the modeling or deployment of CDR (Anderson and Peters, 2016; McLaren, 2020). Furthermore, in GCAM 5.4, demand for DAC and BECCS are quasi-independent; DAC and BECCS technologies cannot compete directly with one another to satisfy overall demand for CDR (but see Sect. 2.4.2 on indirect competition).

GCAM-CDR offers a much wider range of CDR policy options, including the ability for users to mimic the approaches

used in GCAM 5.4. We describe those options below.

### 2.4.1. Paying for CDR by different mechanisms

In GCAM 5.4, negative emissions receive a subsidy equivalent to the carbon tax, as explained above. For example, if technologies face a tax of \$100 per ton on carbon emissions, a technology that removes a ton of carbon from the atmosphere will receive a subsidy of \$100. Since carbon prices in GCAM can easily reach hundreds or even thousands of dollars—well above the projected

long-run costs of CDR—this implies that the CDR industry is modeled as enjoying large economic rents (meaning, roughly, profits in excess of what is required to make the industry economically viable in a market system).

While GCAM-CDR can mimic that behavior, it also offers an alternative at which CDR technologies are paid at market rates, based on the cost of providing CDR. (Optionally, this alternative mechanism can apply to BECCS technologies as well as the new CDR technologies, in which case they receive a market-based payment for CDR services that is just enough to induce

them to supply the amount of CDR demanded of them at that price. See Sect. 2.4.2.) This mirrors the way that prices of other goods and services in GCAM are modeled. Including this option allows users to model situations in which market competition for CDR services or successful cost discovery by policymakers drives prices down to eliminate economic rents. This is particularly important because the high rents are partially responsible for the scale and rapidity of BECCS adoption in deep mitigation scenarios in GCAM.

Users can also apply the full suite of other price manipulation mechanisms in GCAM (e.g., fixed taxes, subsidies, portfolio standards, etc.) to CDR technologies in GCAM-CDR.





### 2.4.2. Integrating BECCS into the CDR market

In GCAM 5.4, BECCS and DAC do not directly compete to satisfy demand for CDR. GCAM contains several BECCS technologies

in different parts of the energy system, mainly in the electricity sector and the refining sector. In GCAM 5.4, the amount of carbon removed via BECCS is determined by the amount of energy demanded from each of these BECCS technologies. That, in turn, depends on the cost of those technologies relative to their competitors, after accounting for the value of any taxes and subsidies to the various technologies. This means that in GCAM 5.4, the amount of CDR from BECCS and DAC, respectively, are determined somewhat independently from each other. DAC deployment affects the amount of BECCS only insofar as it changes the carbon

price. This interaction occurs because changes to the carbon price change the after-tax prices of BECCS technologies and their competitors. Conversely, the amount of BECCS has a similarly indirect effect on the amount of DAC deployed: deploying BECCS can reduce the carbon price, and in general, deployment of DAC is an increasing function of the carbon price, bounded at the top by the exogenously specific maximum demand. In situations where neither BECCS nor DAC influence the carbon price (e.g., in fixed-tax scenarios), they do not compete at all.

GCAM-CDR offers users a simple way to allow BECCS technologies to compete more directly with forms of CDR. If a user enables BECCS integration into the CDR market, BECCS technologies compete directly with other CDR technologies for a share of the CDR market. This involves a number of changes. First, the carbon price-based subsidy to BECCS is neutralized. Second, the demand for CDR from BECCS is determined in the CDR sector using GCAM's standard algorithm for distributing market share among competing technologies, based on an endogenously calculated price. GCAM's solution algorithm adjusts the

price paid to BECCS technologies for their CDR services until the amount of CDR demanded of them, collectively, equals the amount that they are willing to provide. This price is generally lower than the cost of CDR via other technologies, as BECCS technologies treat it as a subsidy to the re2.venue from producing electricity or other energy carriers. Given the way that GCAM allocates market share between competing technologies, this means that BECCS usually captures a significant share of the CDR market, but not all of it. (A similar approach is used in the case of the fertilizer-substitute implementation of TEW.)

Note that integrating BECCS into the CDR market subjects it to the overall constraint on CDR growth described in Sect. 2.3. This leads to much slower adoption of BECCS than typically occurs in rapid decarbonization scenarios in GCAM 5.4.

### 2.4.3. Interregional trade in CDR

GCAM-CDR allows users to enable trade in CDR services between geopolitical regions. The trade is implemented using the same Armington-style approach that GCAM 5.4 uses for many other commodities, such as corn. A supply sector is created that can draw

from CDR output in any region in a multi-region or global market, and any region in that market can purchase CDR services either directly from its own market or through the multi-region market. This could be interpreted as an international clearinghouse for CDR services, in which demand for CDR in one region (e.g., Canada) can be satisfied by CDR in a different region (e.g., China).

When trade in CDR is enabled, GCAM-CDR assumes zero "home bias" (Zhao et al., 2021), meaning that each region regards "imported" CDR as a perfect substitute for domestic CDR. Users may customize trade preferences by adding non-zero

home bias at the regional level or by adding global, non-price-based preferences for CDR services from some regions over others.

By default, when trade is enabled, each region's share of the global CDR market is weighted according to that region's share of global GHG emissions in a reference scenario. Users can easily modify these weightings, but some unequal weighting is necessary to avoid the implausible outcome that each region would supply very similar amounts of CDR, regardless of their geographical or economic size, industrial capacity, and other relevant factors. Weighting by GHG emissions in the baseline case





captures a variety of key drivers of CDR capacity, including economic output, total industrial capacity, and agricultural land available for enhanced weathering. The amount of CDR actually supplied by each region ultimately depends on the region's weighting and its average cost of CDR.

Enabling trade also makes it possible for users to set demand for CDR at the global level and have GCAM-CDR distribute it across geopolitical regions based on regions' capacity for CDR and the cost of CDR in each region.

**2.4.4. Mechanisms for setting demand for CDR**

In GCAM 5.4, overall demand for CDR equals the quantity of CDR via BECCS induced by a given carbon price plus the demand for DAC, if any, which is based on the carbon price and the exogenously specified maximum demand for DAC. Users cannot easily specify total demand for CDR.

GCAM-CDR 1.0 provides a number of approaches to specify total demand for CDR, as shown in Table 1. The three basic
ways of setting CDR demand are: exogenously specifying a quantity of $CO_2$ to remove; allowing demand for CDR to vary endogenously with the carbon price; and setting CDR equal to some fraction or multiple of the emissions in particular regions and/or sectors. These can be combined with one another (e.g., by setting a base demand at a certain level and then allowing additional demand to be driven by the carbon price).

Users can also configure the model so that "unsatisfied" demand accumulates over time. If the CDR sector is unable to
satisfy demand in a particular period (including historical periods) because of a lack of available technologies or user-imposed constraints on the growth of CDR, that demand can be automatically added to future periods at a user-specified rate.

**Table 1. A summary of CDR policy options available in GCAM-CDR 1.0, along with sample research questions or scenarios that could be run with each type. These options can be combined, including more than one policy in a single scenario.**

| CDR Policy | Description | Sample Use(s) |
|---|---|---|
| Exogenous | CDR demand is set exogenously in each period, either regionally or globally. | Setting separate targets for emissions abatement and CDR (McLaren et al., 2019); modeling quantity-based government procurement of CDR; matching CDR demand from scenarios produced in other models |
| Elastic | CDR demand increases as the carbon price increases, up to an exogenously specified maximum. | Modeling flexible amounts of CDR demand that are assumed to rise endogenously with the stringency of climate policy; scenarios with net-emissions constraints in which a flexible proportion of mitigation comes from CDR |
| Offset | CDR demand is set to a fraction or multiple of gross emissions in each year. Users can configure demand to neutralize emissions from specific regions and/or sectors. Users can configure demand to neutralize emissions $CO_2$ and/or other greenhouse gasses. These do not operate as offsets from the perspective of individual emitters: technologies still pay the carbon price on their emissions. | Using CDR to counterbalance emissions from specific sectors; scenarios with net-zero constraints in which a flexible proportion of mitigation comes from CDR; in combination with an Accumulated Demand policy, scenarios involving the clean-up of historical emissions |
| Accumulated Demand | In combination with any of the above policies, users can set CDR demand to carry over to future periods if any demand goes unsatisfied in the current period. | In combination with some other CDR policy, scenarios in which countries, sectors, or companies commit to cleaning up some quantity or fraction of historical emissions |





## 3 Results

To compare the model behavior of GCAM-CDR and GCAM 5.4 and to demonstrate some of GCAM-CDR's new capabilities, we report results from four sets of scenarios. Like GCAM 5.4, GCAM-CDR produces roughly 2 GB of output from each scenario. So, rather than present results in their entirety, we focus on key variables and summary metrics.

### 3.1. Description of the scenarios

Figure 2 illustrates the four sets of scenarios and the relationships between the various scenarios.

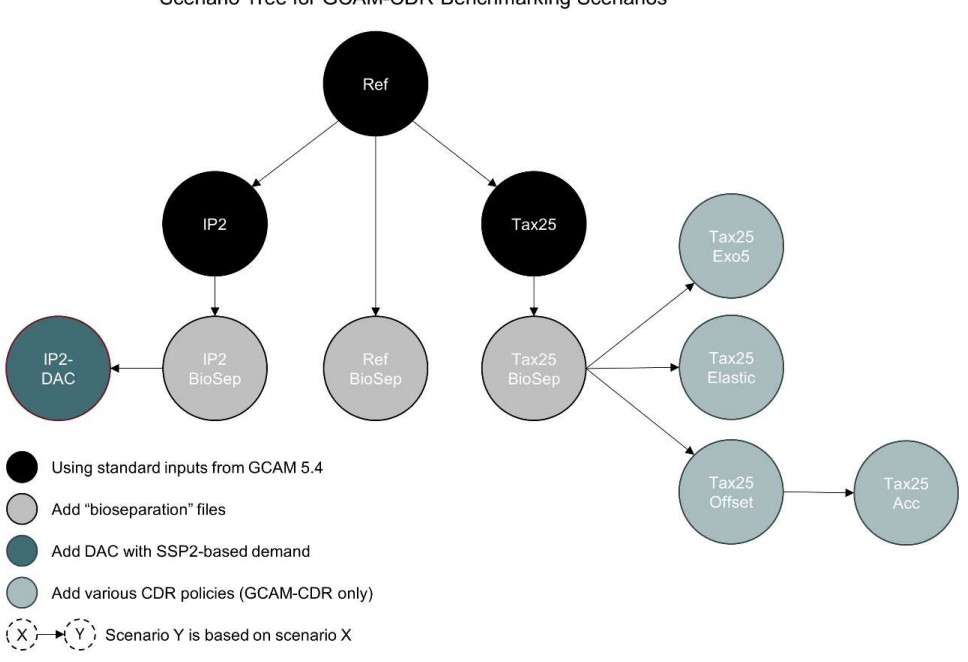

**Figure 2. Scenarios used to compare outputs between GCAM-CDR and GCAM 5.4 and to demonstrate some of GCAM-CDR's capabilities.**

The first set (shown in black in Fig. 2) serves to demonstrate that GCAM-CDR is fully backward compatible with GCAM 5.4 and produces identical outputs when given identical inputs. Each scenario in this set uses default GCAM 5.4 input files and one of three climate policies. The Reference ("Ref") scenario includes no climate policy files, and so features no new climate policy beyond what is already reflected in historical calibration data. A fixed-tax scenario ("Tax25") includes a fixed carbon tax starting at $25/ton in 2025 and rising at 5% per year. A scenario ("IP2") based on Illustrative Pathway 2 from the *Special Report on Global Warming of 1.5°C* (IPCC, 2018; Rogelj et al., 2018) includes a net greenhouse gas-emissions constraint designed to limit warming to 1.5°C at the end of the century. All three scenarios in this set are run in both GCAM 5.4 and GCAM-CDR using identical input files.

The second set (shown in gray in Fig. 2, directly below the first set) serves to quantify the deviations in model behavior caused by the bioseparation files described in Sect. 2.1. The scenarios in this set are identical to the first set, except that each



scenario ("Ref Biosep", "Tax25 Biosep", and "IP2 Biosep") also includes the bioenergy separation files discussed in Section 2. None of these "bioseparation" scenarios involve any technological CDR besides BECCS. All three scenarios in this set are run in both GCAM 5.4 and GCAM-CDR using identical input files.

    The third set (shown in dark blue on the left edge of Fig. 2) serves to demonstrate GCAM-CDR's ability to emulate the behavior of GCAM 5.4 with respect to technological CDR, using GCAM-CDR's technology definition and demand-setting

mechanisms. This set consists of a single scenario ("IP2-DAC") run with slightly different input files in GCAM 5.4 and GCAM-CDR. We first run the IP2 scenario in GCAM 5.4 using GCAM 5.4's implementation of DAC and its SSP2-based assumptions about DAC demand, as developed by Fuhrman et al. (2021). This yields a scenario similar to the high-overshoot scenario described by Fuhrman et al. (2020). We then run the same scenario in GCAM-CDR using GCAM-CDR's implementation of DAC, exogenously setting regional demand for CDR to match demand for DAC in the GCAM 5.4 version of the scenario. For this

scenario, we do not integrate BECCS into the CDR sector, allow trade in CDR between regions, or impose exogenous constraints on the growth of CDR.

    The fourth set (shown in lighter blue on the right side of Fig. 2) serves to demonstrate some of GCAM-CDR's new capabilities. This set is run only in GCAM-CDR because they depend on functionality not available in GCAM 5.4. Specifically, they use GCAM-CDR's full suite of CDR technologies, integrating BECCS into the CDR market, allowing trade in CDR between

regions, and limiting the growth of CDR output to 15% per year. Each scenario uses the same carbon price trajectory as in the Tax25 scenario, but with a different CDR policy from Table 1: an exogenous global demand of 5 GtC (18.33 GtCO$_2$) per year ("Exo5"); an elastic demand that responds endogenously to the carbon price ("Elastic"); an "offset" demand that attempts to offset 100% of CO$_2$ emissions in each period, starting in 2025 ("Offset"); and a scenario that seeks to offset 100% of CO$_2$ emissions in each period from 2005 onward and allows unsatisfied demand from earlier periods to be satisfied later in the century as CDR

capacity grows ("Acc").

### 3.2. Deviations from GCAM 5.4 are small

    The first two sets of scenarios (shown in black and gray, respectively, in the center of Fig. 2) quantify the deviations introduced by GCAM-CDR's base configuration without CDR. The overall deviations in primary energy by source are quantified in Fig. 3, using the taxicab distance metric developed by Krey and Riahi (2013). This distance metric, normalizes deviations between models

or scenarios to the overall scale of primary energy consumption, has been used to measure intermodal consistency in various IAM studies (Clarke et al., 2012; Iyer et al., 2019; Cohen et al., 2021).



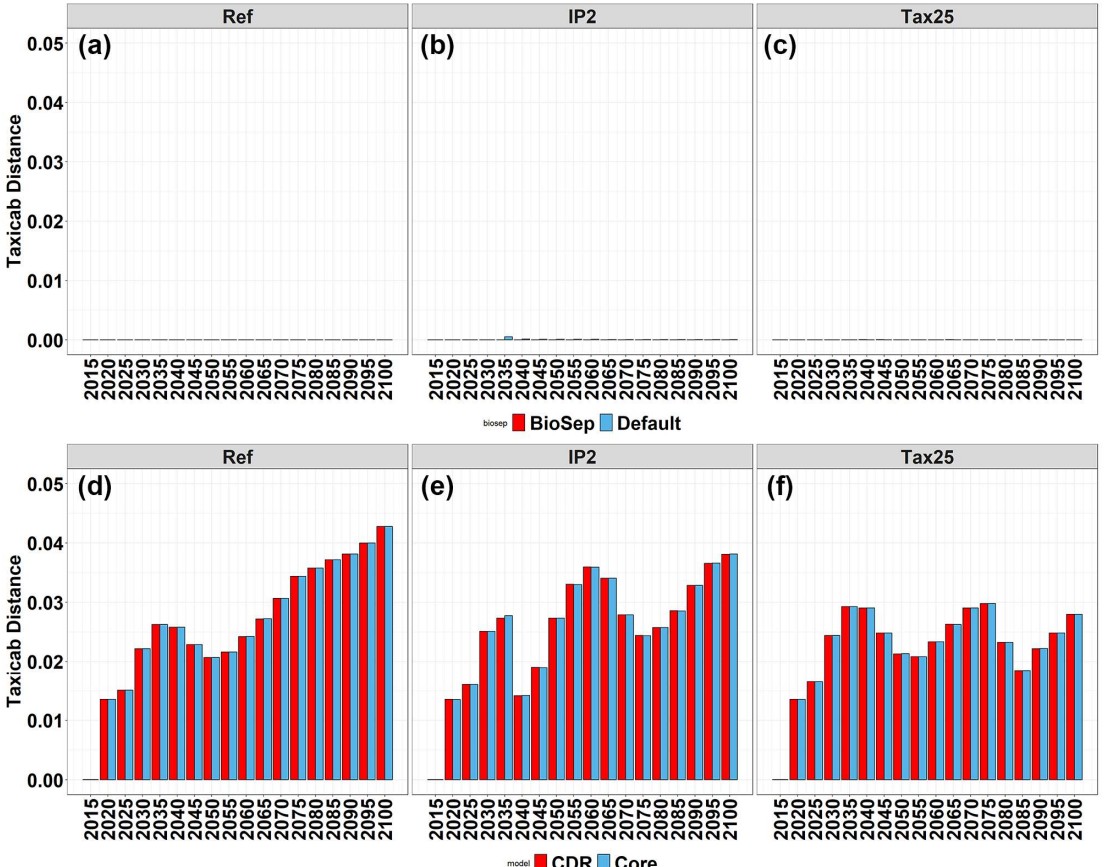

**Figure 3. The between-model taxicab distance for primary energy consumption by source in GCAM-CDR 1.0 and GCAM 5.4 using standard GCAM 5.4 inputs in scenarios that include no new climate policy (a), an ambitious carbon constraint (b), and a moderate carbon price (c); plus the within-model taxicab distance between outputs with and without the "bioseparation" files (d,e,f) in scenarios with the same climate policies as in (a,b,c).**

Without the bioseparation files, the models behave virtually identically. We hypothesize that the miniscule differences that do arise result from differences in the rounding of floating point numbers, caused by differences in compiler output between the two executable files, which cause the solution algorithm to follow very slightly different paths. When the bioseparation files are included, both models deviate noticeably from the "unseparated" scenarios. The deviations differ qualitatively depending on the climate policy, but they remain small overall. Using the taxicab metric, the overall "distance" between equivalent scenarios is <0.05 in every period and usually <0.03. The taxicab distance between the unseparated and bioseparated scenario in each period is virtually identical across models, implying very similar model behavior in response to the bioseparation files.

The bioseparation files do cause shifts in energy sources that are non-negligible in absolute terms, though, as shown in Fig. 4. Specifically, the bioseparation files induce a noticeable shift in consumption of some energy sources, especially oil and biomass, but the shifts are qualitatively different depending on the scenario's climate policy. This is most prominent in the refined liquids sector. When carbon prices are low or nonexistent, as in Tax25 and Ref scenarios, refined petroleum products are partially



displaced by additional bioliquids. (For a sense of scale, the increase in bioliquids in 2100 in the Ref_BioSep scenario amounts to a change in feedstock for 10.9% of global refined liquids production.) When carbon prices are high, as in most of the IP2 scenario or the end of the Tax25 scenario, the reverse happens: refined petroleum products increase and bioliquids decrease, relative to the equivalent unseparated scenarios. This is partly driven by the bioseparation file's exclusion of bioliquids from the industrial feedstocks sector, which can consume large quantities of bioliquids when the carbon price is high enough. (GCAM counts the

creation of industrial feedstocks from carbonaceous liquids as a form of $CO_2$ sequestration. GCAM 5.4 normally constrains that sector to use mainly petroleum-based fuels, but at high enough carbon prices, large amounts of bioliquids are also used for feedstocks because they receive a subsidy for doing so. Due to the highly variable duration of sequestration in industrial feedstocks, GCAM-CDR excludes bioliquids from the industrial feedstocks sector altogether to prevent dubious claims of CDR in that sector, resulting in noticeably different behavior in that sector.)

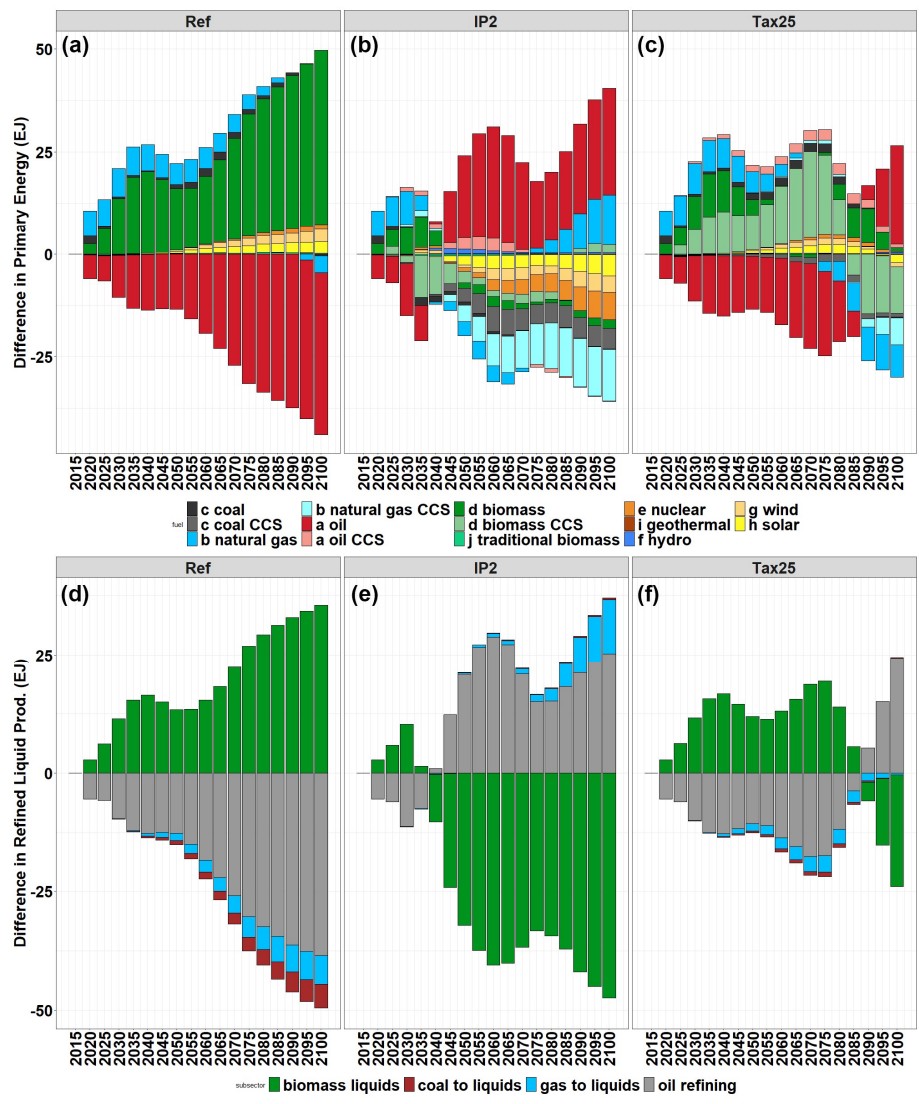


Figure 4. Absolute difference in primary energy sources (a,b,c) and refined liquids inputs (d,e,f) when adding "bioseparation" files to scenarios with no new climate policy (a,d), an ambitious carbon constraint (b,e), or a moderate carbon price (c,f).



### 3.3. Similar CDR policies produce similar results in both models

Scenario IP2-DAC demonstrates GCAM-CDR's ability to emulate the behavior of DAC and BECCS in GCAM 5.4, despite the

minor differences in the implementation of DAC between the two models and the major differences in the way users specify CDR

demand. Fig. 5 shows virtually identical sectoral emissions trajectories in both models' implementation of the scenario.

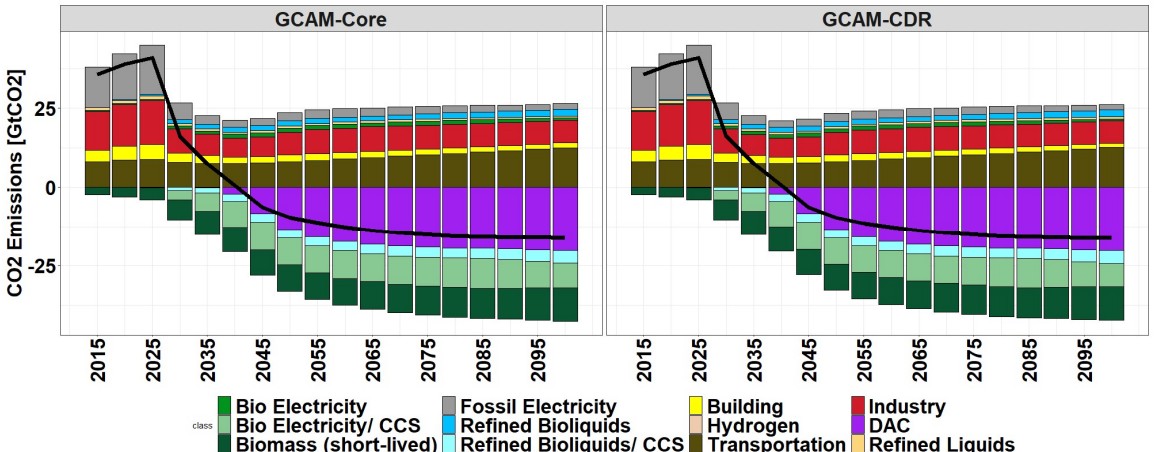

**Figure 5. A comparison of GCAM-CDR and GCAM 5.4 with DAC and BECCS using identical demand for DAC. The dark green negative**
**emissions, labeled "Biomass (short-lived)," refer to $CO_2$ that is captured from the atmosphere during the growth of biomass but re-**
**emitted in another sector (e.g., Transportation) shortly after harvesting. By contrast, negative emissions from "Bio Electricity/CCS" and**
**"Refined Liquids/CCS" refer to $CO_2$ that is captured from the atmosphere during the growth of biomass and then permanently**
**sequestered in geological formations.**

### 3.4. Different CDR policies produce very different outcomes

The last set of scenarios (shown in Orange in Fig. 2) demonstrate that, keeping all else constant, configuring CDR policies

differently leads to very different outcomes. Each scenario is derived from the CTax25_BioSep scenario. Each scenario therefore

includes the same trajectory of carbon taxes, beginning at $25/ton in 2025 and rising at 5% per year. The scenarios differ from

CTax25_BioSep in two ways: all scenarios include GCAM-CDR's full suite of CDR technologies, along with interregional trade

in CDR services; and each scenario includes a different CDR policy from Table 1. (See Sect. 2.4.4.) Note that all four scenarios

contain an identical, exogenous constraint on the growth of CDR, which limits global growth in CDR output to 15% per year.

Fig. 6 depicts the CDR output by technology, as well as the impact of these different policies on the $CO_2$ concentrations

and global mean temperature anomaly.





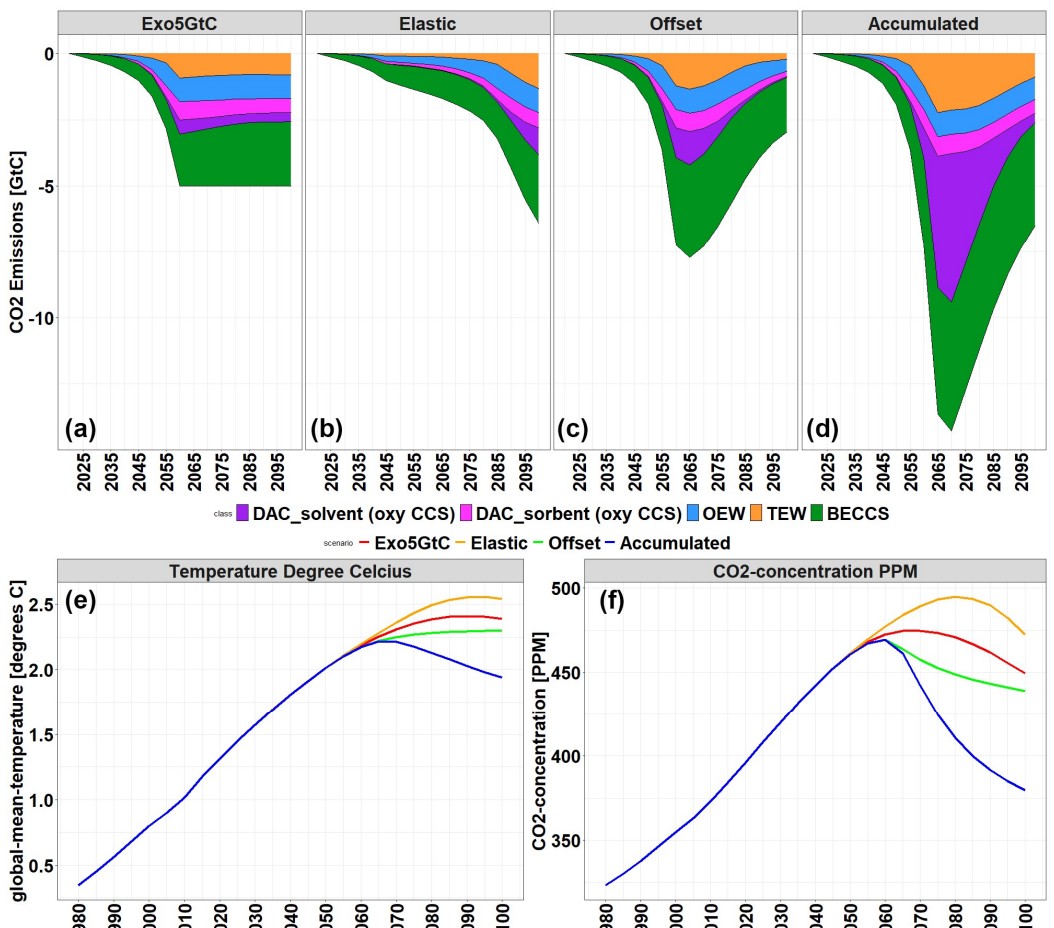

**Figure 6. Results from GCAM-CDR with different CDR policies. Different policies produce significantly different amounts of CDR once the exogenous constraint on CDR growth has ceased to be binding on CDR output. (a) "Exo5GtC" imposes a global CDR demand of 5GtC (18.33 GtCO₂) per year. (b)"Elastic" sets CDR demand to rise with the carbon price, reaching 6.39 GtC (23.43 GtCO₂) globally in 2100. (c) "Offset" sets CDR demand to attempt to offset regional CO₂ emissions in each period after 2020, subject to CDR capacity constraints. (d) "Accumulated" attempts to offset all CO₂ emissions from 2005 onward, with unsatisfied demand from earlier in the century being satisfied later in the century as CDR capacity increases. Different CDR policies lead to very different temperature (e) and CO₂ concentration (f) trajectories later in the century, with end-of-century values differing by more than 0.5°C and roughly 100 ppm between the highest and lowest scenarios.**

## 4 Discussion and conclusions

GCAM-CDR builds on GCAM 5.4 in two main ways: by introducing some additional CDR technologies and by introducing a wider range of more flexible CDR-related policies. The former represents an incremental improvement, especially given the overlap between the CDR technologies represented in GCAM-CDR and those introduced in GCAM 5.4. The latter represents a significant improvement because it enables users to study a much broader range of possible climate policies.

Future research directions using GCAM-CDR are likely to include analyses of both possible CDR policies and various CDR technologies. With respect to the former, questions abound about the impact of different kinds of CDR policies on outcomes related to climate, socioeconomics, health, energy and materials usage, sustainability, and the equitable division of responsibility for CDR. With respect to the latter, one likely avenue for further research includes the introduction of additional CDR technologies,

1

especially those that use different energy or material inputs, such as DAC powered entirely by electricity, and those with significant potential co-benefits, such as soil carbon sequestration, biochar, carbon utilization, or carbon-negative hydrogen.

At the same time, IAMs should not be mistaken for crystal balls, and GCAM-CDR is no exception. Given the large uncertainties around key parameters with respect to CDR, as well as harder-to-model uncertainties surrounding sociotechnical systems for CDR, model results must be interpreted thoughtfully when trying to explore these issues. As the saying goes, "models are for insights, not numbers." Although IAMs report projections in exquisitely quantitative detail, these numbers should not be taken too literally. To illustrate this point, consider the relative output of high-heat DAC and low-heat DAC depicted in Fig. 6. Competition between CDR technologies in GCAM-CDR is based mainly on cost. While cost parameters for these technologies

reflect the best current estimates, the cost of DAC technologies several decades from now remains highly uncertain. It is also worth noting that GCAM 5.4 (and therefore GCAM-CDR) projects higher per-unit energy costs than are used in many of the long-term cost estimates in the scientific literature, which changes the competitiveness of more energy-intensive forms of CDR relative to more capital-intensive forms. GCAM-CDR's projections should not, therefore, be taken as definitive statements about the relative cost or promise of any technology. Instead, they should be examined for important dynamics. For instance, low-heat DAC in

GCAM-CDR uses waste heat from thermal power plants and industrial sources, rather than producing heat from natural gas or other sources. This gives it an advantage in scenarios with lower overall CDR demand but a distinct disadvantage in scenarios with very high overall CDR demand, such as the Accumulated scenario, in which there simply is not enough waste heat available in GCAM-CDR to power vast quantities of low-heat DAC. This lesson applies broadly to CDR approaches that tap into waste streams, such as DAC using curtailed renewable electricity or enhanced weathering using alkaline wastes. It also implies a complicated

relationship between scalability, inputs, and the projected long-term demand for CDR. As always, the question is what can be legitimately inferred from the model's behavior and what new questions are raised by that behavior.

**Code Availability**

GCAM-CDR is an open source model. The version of GCAM-CDR described in this paper is archived on GitHub and Zenodo ( https://doi.org/10.5281/zenodo.6497952, Morrow et al., 2022). The complete model and a user guide are available at

https://github.com/icrlp/gcam-cdr. Configuration files for the scenarios reported in this paper are available in the Supplemental Information.

**Author Contributions**

DRM, RA, and GG all participated in the development of GCAM-CDR. DRM led the writing of the paper with contributions from RA and GG. Simulations in the paper were designed by DRM, RA, and GG; they were conducted and analyzed by RA. Figures

were produced by DRM and RA.

**Competing interests**

The authors declare that they have no conflict of interest.

**Acknowledgments**

The development of GCAM-CDR and the writing of this paper were supported by a grant from the Alfred P. Sloan foundation.



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
