# Peer review of "GCAM-CDR v1.0: Enhancing the Representation of Carbon Dioxide Removal Technologies and Policies in an Integrated Assessment Model"

_Geoscientific Model Development, 2022_

## Author Response (AR1)

| \multicolumn | | |
|---|---|---|
| **SUMMARY OF COMMENTS, RESPONSES & CHANGES FOR GMD-2022-125** | | |

| RV'ER | REVIEWER'S COMMENT | AUTHORS' RESPONSE | CHANGES TO MS |
|---|---|---|---|
| 1 | As a general comment, it would be very useful to report also costs of the different technologies described in the manuscript. | We have deliberately de-emphasized the costs in the paper, for reasons that we now explain in the Supplemental Information. The SI now contains a lengthy discussion of costs. This includes an illustrative example of the cost of different technologies in a particular geographic region for a particular year. As the SI explains, the reason we de-emphasize the costs is because GCAM's high energy prices inflate the per-tonne costs of CDR technologies, relative to the CDR literature. Because we do not think this reflects a reliable forecast of the costs of CDR, we do not want to give the impression that our model implies that CDR is more expensive than people are predicting. What matters for the model, as we now explain in the SI, is the relative costs of the CDR technologies, which are so heavily influenced by energy costs (which are, in turn, variable across scenarios) that it is unhelpful to try to summarize CDR costs in the paper. | Added a section in SI about costs, including an illustrative example of the cost of different technologies in a particular geographic region for a particular year. |
| 1 | P4 l98-105: The approach for CDR to compete with a placeholder technology to limit growth is unclear. How is the placeholder technology modelled? How is ensured that this technology is competitive, but doesn't have or produce energy or money? Why is this approach chosen, and not a direct constraint modelling the actual constraints? | We explain this in slightly more detail in the main text, but as the full explanation and rationale is rather technical, we have added a new section in the SI to elaborate on it. In short, we are adapting a technique already used in GCAM 5.4, with some changes to account for GCAM-CDR's different approach to driving demand for CDR. | Minor changes in main text (p. 4, ln 205ff in new MS), plus a new section in the SI. |
| 1 | P4 high-heat DAC: why can only be gas used to generate the high temperatures and not H2? | As we now explain in Section 2.3 of the main text, we are focusing on a subset of the many different CDR technologies in order to illuminate the basic dynamics of having a suite of available CDR technologies and policies. Users can easily add variations of these technologies. | See revisions to Section 2.3 in main text, explaining that we are not trying to be comprehensive in available technologies. |

| SUMMARY OF COMMENTS, RESPONSES & CHANGES FOR GMD-2022-125 | | | |
|---|---|---|---|
| RV'ER | REVIEWER'S COMMENT | AUTHORS' RESPONSE | CHANGES TO MS |
| 1 | Is there a justification for taking the lower estimates of energy requirements from Realmonte? | We use the lower estimates because we expect them to be more accurate than the upper-bound estimates by the time the technology is usually deployed in model. Users can easily adjust those parameters to reflect different levels of optimism. | Explain in the detailed tech descriptions in the SI that rapid improvements in DAC technology make us optimistic that energy requirements will be closer to the lower bound, but that users can easily tweak these parameters. |
| 1 | P4 low-heat DAC: Is there a justification for taking the lower estimates of energy requirements from Realmonte? Why can the low-temperature heat not be provided via electricity as well, e.g. heat pumps? From the conclusion I take that the availability of waste heat is a limitation for this technology. This doesn't seem plausible and could be solved by allowing for other heat sources, which would of course increase the costs for deployment beyond the availability of waste heat. | On the first sub-question, see response to previous question. On the second sub-question, see response to the question before that. With respect to waste heat, one advantage of using something that has a binding constraint is that it helps us examine the dynamics of the CDR sector when some technologies hit those constraints. | No changes to MS other than those made in response to previous queries. |
| 1 | P4 TEW: According to Strefler et al., 1 t basalt binds 0.3 tCO2. If I understand the numbers in the SI correctly, here 1 t C is assumed, which would be a factor of 10 off. | The reviewer is correct, and our parameter was not. We have fixed the parameter, uploaded a corrected version of the model to Zenodo and GitHub, and verified that the changes to the parameter do not affect any of the results reported in the paper. | No changes to MS, but we uploaded changes to the model to Zenodo and GitHub. |
| 1 | P5 ocean liming: Again, the numbers seem to be on the optimistic side of the range given in Renforth et al. | Renforth gives specific numbers for the particular technology that we're using (Oxyflash calcination with CCS). This technology is currently being developed for commercial use by Origen. | No change. |

| SUMMARY OF COMMENTS, RESPONSES & CHANGES FOR GMD-2022-125 | | | |
|---|---|---|---|
| RV'ER | REVIEWER'S COMMENT | AUTHORS' RESPONSE | CHANGES TO MS |
| 1 | Also, it doesn't seem plausible that the availability of cargo ships limits the capacity of ocean liming. Building dedicated ships would certainly be possible, though this would increase costs. | Again, we're starting with a limited set of technologies, and we chose to model ocean liming from existing cargo vessels. This also has the advantage of being able to model a steeply upward-sloping supply curve related to the fact that large numbers of cargo ships frequently sail "ballast legs" in which they often taken unprofitable cargo at a steep discount rather than sail empty. | We have now explained this in the main text (p. 5, ln 171-172) and SI and explained how users can easily introduce a variation of ocean liming that uses a purpose-built fleet and escapes this constraint. |
| 1 | P6 l192: Depending on the climate target, [the independence of DAC and BECCS] seems implausible. Why would there be separate targets for DAC and BECCS, if the main output provided at least by DAC (i.e. CDR) could also be fulfilled via BECCS? | We actually agree with the reviewer here, who has misinterpreted our description of GCAM 5.4 as an endorsement of its approach, rather than as pointing out a limitation of it. | We have clarified in the MS (p. 7, ln 234-236) that we see this as a limitation and that the default configuration in GCAM-CDR is to have BECCS compete directly with DAC, etc. |
| 1 | P6 l202ff: This mechanism [for paying CDR techhnologies] is confusing. CDR options like DAC are constrained mainly by energy supply, which could be increased, driving prices up. So if DAC is always paid at market rates, how is the demand limited? | Demand for DAC etc. is set (and therefore limited) as described in Section 2.4.4. | We have edited the text in this section (2.4.1) to clarify that this mirrors the way prices are usually determined in GCAM and that its main purpose is to eliminate the economic rents implied by the carbon-price-based approach. The mechanism for setting (and limiting) demand for CDR is addressed later in the text. |
| 1 | P7 l215: The energy could also be provided by bioenergy technologies without CCS. What is the incentive for using BECCS instead? | GCAM includes both BECCS and bioenergy without CCS. We mentioned only BECCS in the original MS because that is the form of bioenergy that competes (indirectly) with DAC in GCAM 5.4. | We have edited the text in section 2.4.2 (first para.) to clarify that there are other forms of bioenergy available in GCAM 5.4, and that BECCS is incentivized by carbon subsidies. |
| 1 | P7 l226: Why is the default case to have BECCS separated from the CDR market? | This is the default case (in one sense of default) only because of the internal structure of GCAM, which requires users to provide specific inputs to integrate the two markets. But the default configuration for GCAM-CDR includes those specific inputs, so in a different sense of 'default', the default case is to integrate the two markets. | We have added a clause on p 7, ln 236-237 to clarify that the default in the GCAM-CDR configuration is to integrate the markets. |

| | SUMMARY OF COMMENTS, RESPONSES & CHANGES FOR GMD-2022-125 | | |
|---|---|---|---|
| RV'ER | REVIEWER'S COMMENT | AUTHORS' RESPONSE | CHANGES TO MS |
| 1 | P7 l246ff: This [way of distributing CDR across regions] is an arbitrary choice. CDR could also be distributed according to the economic efficient solution, or according to other equity schemes. Please explain the reasoning behind this choice. | It seems we did not clearly explain ourselves here. Weighting by GHG production (to which the reviewer objects) is just an initial weighting, and that when interregional trade in CDR is enabled, GCAM-CDR allocates CDR across regions based on economic efficiency, weighted by GHG production. | We have edited paragraph around ln 260 to clarify how this works. |
| 1 | P12 l342: I don't see why bioliquids should not be used as feedstock. It requires a proper accounting of the lifetime of these feedstocks, but then also the use of fossil fuels does as this would also lead to emissions. | In principle, we agree, but GCAM is not currently equipped to do this. Given the model's current capabilities, prohibiting bioliquid feedstocks avoids some unrealistic model behaviors. This is another potential avenue for futher research and development | We have added a sentence explaining our decision on ln 362ff. |
| 1 | P13 l365: Why [limit growth in CDR to] 15% per year? This is an arbitrary choice, please explain the reason behind this number. | The answer here is a long one, which we now discuss in detail in the new section in the SI on constraining the growth of CDR. It is worth emphasizing, in the context of the figures of the main text, that not much would change with a different growth rate: the output of CDR would rise more or less sharply over the first half of the century, and temperatures and CO2 concentrations would be slightly higher or lower, but the main differences between scenarios are driven by difference in final demand for CDR, not by the choice of growth rate. | See the new section in the SI. |
| 1 | P2 l68: I assume you mean GCAM-CDR here and not GCAM 5.4 | Correct. | Fixed typo. |
| 1 | P7 l232: typo in "revenues". | Correct. | Fixed typo. |

| SUMMARY OF COMMENTS, RESPONSES & CHANGES FOR GMD-2022-125 | | | |
|---|---|---|---|
| RV'ER | REVIEWER'S COMMENT | AUTHORS' RESPONSE | CHANGES TO MS |
| 2 | I agree with the first referee that reporting of the numerical costs and performance in the main body of the manuscript would be useful. While I see this is done in the Supplementary Information, it would be helpful to have in the main manuscript and reported in units that are more intuitive (e.g., GJ/tCO2), and include the levelized non-fuel cost assumptions as well (e.g., 2020 USD/tCO2) as the model results are highly sensitive to both parameters. | See our response to Reviewer 1's first comment (row 3 in this spreadsheet). We take the point about reporting parameters in more intuitive units. | In addition to the changes described in Row 3 of this spreadsheet, we have recalculated the energy I/O coefficients in terms of GJ/tCO2 wherever they appear (namely, in the figures and table in the SI). |
| 2 | 1. 4 L-121. GCAM 5.4 represents a sorbent-based DAC process wherein the low-temperature heat is assumed to be supplied by an electric heat pump with an assumed coefficient of performance and thus does not require any natural gas input. The model also includes representation of a high-temperature DAC process which again uses only electricity to provide the high-temperature heat requirement. This sentence should be clarified to avoid implying only the natural gas-based process is represented in the model. | The reviewer is correct. | We have added and/or edited sentences at the end of sections 2.3.1 and 2.3.2, respectively, to explain more clearly what kinds of DAC are included in GCAM 5.4. |
| 2 | On a related note, in the "DAC.xml" input file, and in Figure 6, the naming "DAC_sorbent (oxy CCS)" seems to imply oxy-fuel combustion, which is not used in solid sorbent-based DAC processes. | This was a labeling error on our part, resulting from an overzeaolous find-and-replace in the XML. We are grateful to the reviewer for catching it. | We have corrected the XML input file in the new version of the model uploaded to GitHub and Zenodo. We have corrected the legend on Fig 6. |

| SUMMARY OF COMMENTS, RESPONSES & CHANGES FOR GMD-2022-125 | | | |
|---|---|---|---|
| RV'ER | REVIEWER'S COMMENT | AUTHORS' RESPONSE | CHANGES TO MS |
| 2 | In the waste_heat_endogenous.xml file, the source and derivation of the "output-ratio" parameter defining the amount of waste heat produced per unit of e.g., thermal power generation or industrial energy use should be provided for each of the technologies for which it is defined. Same for the 0.42 price at which 100% of the maximum waste heat available is provided. | The answer here is also a long one, but in short, waste heat availability is calculated as a fraction of the difference between energy inputs and energy outputs for each of these technologies, with the cost set at a rate that brings the overall cost into line with independent projections. | This is now explained in the technology description in the SI. |
| 2 | 1. 4 L-125. TEW: The assumptions regarding rock comminution particle size and upper or lower bound estimate from Streffler et al., 2018 used to parametrize the electricity input parameter should be provided in the SI. | We are assuming comminution to 10 nm, which is within the ranges found in the literature on enhanced weathering. We experimented with including a variety of particle sizes in the model but found 10 nm to strike a good balance (given other model parameters) between efficiency and cost. | We explain our choice of rock particle size in the SI technology description |
| 2 | 1. 5 L-150. OEW: Why is the shipping input a by-product of international shipping, rather than having this service as a direct input? Distributing the limestone or other alkalinity over the ocean surface "consumes" some amount of tonne-km of international shipping capacity. This would seem to make direct rather than co-product consumption of this service a more appropriate modeling approach. | There are two main reasons for this choice. The first is that, unlike using shipping as a direct input, the byproduct approach enables to capture the ability of international shipping industry to use "ballast runs" (in which cargo ships sail empty or with unprofitable cargo) to distribute lime, such that the cost of shipping for ocean liming starts at a fraction of actual cost and rises with demand. The second is that this gives us a way to impose endogenous limits on ocean liming, which enables the model to explore some interesting questions. But users can fairly easily add a variation that uses shipping as a direct input. | We have edited Section 2.3.4 in the SI to explain this in slightly more detail. |